# The Comparison of Lifestyles, Mental Risks, and Physical Indices among Individuals with Ultra-High Risk of Psychosis, Patients with Schizophrenia, and the General Public

**DOI:** 10.3390/bs14050395

**Published:** 2024-05-09

**Authors:** Ching-Lun Tsai, Chi-Ming Chu, Hsien-Yuan Lane, Shiah-Lian Chen, Cheng-Hao Tu, Wei-Fen Ma

**Affiliations:** 1Department of Research and Development, BIOZOE Co., Ltd., Taichung 404, Taiwan; chris.tsai70@gmail.com; 2School of Public Health, National Defense Medical Center, Taipei 114, Taiwan; cm.chu.tw@gmail.com; 3Big Data Research Center, Fu-Jen Catholic University, New Taipei City 242, Taiwan; 4Department of Public Health, School of Public Health, China Medical University, Taichung 404, Taiwan; 5Department of Public Health, Kaohsiung Medical University, Kaohsiung 807, Taiwan; 6Graduate Institute of Biomedical Sciences, China Medical University, Taichung 404, Taiwan; hylane@gmail.com; 7Department of Psychiatry, China Medical University Hospital, Taichung 404, Taiwan; 8Department of Psychology, College of Medical and Health Sciences, Asia University, Taichung 413, Taiwan; 9Department of Nursing, National Taichung University of Science and Technology, Taichung 403, Taiwan; shiah90@nutc.edu.tw; 10Graduate Institute of Acupuncture Science, China Medical University, Taichung 404, Taiwan; lordowen@mail.cmu.edu.tw; 11School of Nursing, China Medical University, Taichung 404, Taiwan; 12Department of Nursing, China Medical University Hospital, Taichung 404, Taiwan

**Keywords:** health needs, lifestyles, mental risks, schizophrenia, ultra-high risk, at-risk mental state

## Abstract

(1) Background: Early interventions may effectively reduce the risk of mental disorders in individuals with ultra-high risk. Specifying the health needs of individuals with ultra-high risk is crucial before the implementation of successful early intervention. This study aimed to explore the differences in lifestyles, mental risks, and physical indices among individuals with ultra-high risk, patients with schizophrenia, and healthy subjects. (2) Methods: A cross-section design applying seven questionnaires with physical examinations for 144 participants aged 13–45 years old was conducted in this study. The questionnaires included one about personal data, four on mental risks, and two for lifestyles. (3) Results: The individuals with ultra-high risk scored similarly in many dimensions as the patients with schizophrenia, but they displayed lower positive symptoms, lower negative symptoms, lower prodromal symptoms, higher interpersonal deficits, lower nutrition intake, and higher levels of exercise than the patients with schizophrenia. Female individuals with ultra-high risk had lower self-esteem, higher positive symptoms, lower nutrition intake, and higher exercise levels than male ones. (4) Conclusions: The study pinpointed specific health needs with interpersonal deficits, nutrition intake, and physical activity for the individuals with ultra-high risk. Future interventions targeted on improving social function, dietary pattern, and exercise will be beneficial.

## 1. Introduction

Applying early intervention for people with mental health problems as early as possible can promote independence, recovery, and self-sufficiency, as well as facilitate employment opportunities and social activities [1,2]. It helps mitigate the overall impairment of the patient and prevent the deterioration of the disease [3]. Early intervention may prevent health damage and delay illness onset for people at risk of prodromal psychosis [3,4,5]. Therefore, distinguishing the characteristics and the special health needs of prodromal psychosis plays a crucial role in helping individuals at risk of developing mental illness.

Schizophrenia is a brain disorder that seriously influences an individual’s cognitive, emotional, social, and occupational functions [6]. Prior to the onset of psychosis, an individual may undergo a phase of minor symptoms and deteriorating functions. This phase is called the “prodromal stage of psychosis” [7,8]. Pioneering studies focused on pre-psychotic signs and symptoms as high-risk criteria to examine prodromal symptoms as “ultra-high risk” (UHR), including recent history of attenuated psychotic symptoms, brief limited intermittent psychotic symptoms, or presumed genetic vulnerability [3,9,10]. Interventions for individuals with UHR typically involve pharmacotherapy (e.g., risperidone, olanzapine, dietary supplements) and psychotherapy (e.g., family therapy, cognitive behavioral therapy), either separately or in combination [3]. These approaches aim to reduce symptoms and prevent the onset of full psychosis [4].

However, for an effective early intervention, it is essential to identify and characterize the specific health needs of those with UHR and how these needs are different from those of patients with schizophrenia and the general public. In a recent study comparing cognitive domains, the individuals with UHR had significantly impaired speed of processing, working memory, and verbal learning, reasoning, and problem solving as compared to the general population. On the other hand, they outperformed the schizophrenia group with illness duration of >3 years in terms of speed of processing and working memory [11].

A study found that UHR individuals and patients with a first episode of schizophrenia showed a similar pattern of neuropsychological dysfunction, albeit with less severity. Further, the individuals with UHR were impaired in attention, working memory, executive function, and verbal memory compared to the healthy people [12]. In the context of family functionality and social support, individuals with UHR had decreased family support and family cohesion relative to the healthy controls but had greater perceived social support than the patients with first-episode schizophrenia [13]. These previous studies highlighted the differences in cognitive and social dimensions between the three groups, but very little is known about their differences in lifestyles.

Regarding lifestyles, previous research has shown that individuals exhibiting psychotic symptoms are more likely to engage in unhealthy lifestyle behaviors, including physical inactivity [9,14,15,16] and poor dietary habits [15,16]. Other studies also highlighted the importance of health-promoting lifestyles in individuals with psychiatric disorders due to their psychotic symptoms influencing their total health and health behaviors [17,18]. However, very few studies were conducted for three-group comparisons in health-promoting lifestyles. In addition, it is also vital to prioritize the most needed aspects in order to refine the care plan to make it suitable for UHR individuals. Therefore, the aims of the study were to understand the differences in psychological health in terms of mental risks, lifestyles in terms of health promotion and quality of life, and physical health in terms of exercise level, physiological measurements, and blood serum tests among healthy persons, patients with schizophrenia, and individuals with UHR.

## 2. Materials and Methods

### 2.1. Study Design

The cross-section design used a purposive sample of those aged from 13 to 45 years old in three groups. The study data were collected from March 2020 to November 2023.

### 2.2. Study Participants

The individuals with UHR (the R group) and patients with schizophrenia (the S group) were referred by psychiatrists from a psychiatric clinic, and healthy control individuals (the C group) were recruited from the general public in the same community.

The general inclusion criteria included (1) age 13–45; (2) good physical health determined by complete physical examination and laboratory tests; and (3) capacity and willingness to give written informed consent. The exclusion criteria were (1) concurrent medical or neurological illness which may possibly influence cognition; (2) premorbid IQ < 70; and (3) current abuse of alcohol or drugs or more than a 5-year history of alcohol or drug abuse. The additional inclusion criteria for the R group included individuals meeting the criteria of Prodromal Syndrome [8,19,20] without a history of comorbid substance use disorder. The exclusion criteria were DSM-5 criteria for schizophrenia, mood disorders, or other psychotic disorders.

An additional inclusion criterion for the S group was inpatients or outpatients meeting Diagnostic and Statistical Manual of Mental Disorders (DSM)-5 [6] criteria for schizophrenia without a history of comorbid substance use disorder. The C group comprised healthy subjects recruited by matching their age and gender with participants in the previous two groups. The exclusion criteria included any diagnosis for schizophrenia, mood disorders, or other psychotic disorders according to DSM-5 criteria.

### 2.3. Instruments

#### 2.3.1. Demographic Inventory

A demographic inventory included gender, age, education years, marital status, religion, previous mental health history, and family history of mental health.

#### 2.3.2. Chinese Version of Schizotypal Personality Questionnaire-Brief (CSPQ-B)

The CSPQ-B was translated from English into Chinese [21,22]. The questionnaire contains 22 self-report true-or-false questions based on a three-factor structure. There are eight items belonging to cognitive–perceptual deficits, eight to interpersonal deficits, and six to disorganization. A higher score indicates greater deficit severity. The best cutoff score was found to be 17 with a sensitivity of 80.0% and a specificity of 85.9% [21].

#### 2.3.3. Brief Self-Report Questionnaire for Screening Putative Pre-Psychotic States (BQSPS)

BQSPS [23] incorporated fifteen true-or-false questions based on a four-factor structure: five questions were related to interpersonal difficulty/social anxiety, four to subthreshold psychotic-like experiences, three to self-depreciation, and three to negative symptoms. A higher score indicates a more severe degree of deficit. BQSPS reported a sensitivity of 0.78 and a specificity of 0.71 [24]. A screened individual answering “true” to more than eight questions or to more than three questions including any of question 1, 2, or 15 was deemed to be in a putative pre-psychotic state [23,24].

#### 2.3.4. Scale of Prodromal Symptoms (SOPS)

SOPS [25] is a 19-item questionnaire created for identifying individuals having early signs of psychosis. It has four subscales: five items in positive symptoms, six items in negative symptoms, four items in disorganization, and four items in general symptoms. A higher score indicates more severe psychiatric symptoms. SOPS had Cronbach’s alpha indices of 0.88 in the recruitment phase and 0.95 one year later [25].

#### 2.3.5. Chinese Mandarin State and Trait Anxiety Inventory Form Y (CMSTAI-Y)

The CMSTAI-Y was translated from the English version [26,27]. It has 20 items each in state and trait anxiety. A higher score indicates a higher degree of anxiety. The range score for a high level of anxiety was 60–80 [27]. The two-week test–retest reliabilities for CMSTAI-Y were 0.76–0.91. The Cronbach’s αs for internal consistency of state and trait anxiety were 0.91 and 0.92, respectively [26].

#### 2.3.6. Health-Promoting Lifestyle Profile—Short (HPLP-S)

The HPLP-S was revised and translated [28] from the original HPLP [29]. It has 24 items in six subscales: stress management, self-actualization, health responsibility, interpersonal support, exercise, and nutrition. A higher score indicates better performance on health-promoting lifestyle behaviors. The internal consistency coefficient of the total scale was 0.90 and ranged from 0.63 to 0.79 for the subscales [28].

#### 2.3.7. Three-Month Physical Activity Checklist (3MPAC)

The 3MPAC is a self-report questionnaire with 18 items measuring physical activity levels in the past three months for adults with psychiatric disorders [30]. The test–retest reliability ranged from 0.71 to 0.86. The criterion validity testing values with a 7-Day Physical Activity Recall interview for light, moderate, and heavy exercises were r = 0.47, 0.64, and 0.73, respectively. The cross-sample testing was *χ*^2^ = 21.98, *p* < 0.001 [30].

#### 2.3.8. Physiological Index and Blood Serum Examination

The body mass index (BMI), systolic blood pressure (BP), diastolic BP, and waist/hip ratio were measured using non-invasive approaches. The glucose (AC), triglyceride, and HDL were detected in blood serum.

### 2.4. Study Procedure and Ethical Considerations

The ethics approval of this study was granted through the Institutional Review Board of China Medical University Hospital, Taiwan (CMUH106-REC3-158). After obtaining approval from the institutional review board and emphasizing our respect for participants’ autonomy and readiness to protect their privacy, study participants returned their consent. The purpose and procedure of the study were clearly explained to the participants. They were entitled to withdraw from the study at any time with no need to provide a reason, and the withdrawal did not affect any of their rights. No research data were made available to any third party, organization, or institution.

### 2.5. Data Analysis

SPSS for Windows (version 22.0) was used for data analysis. The descriptive analysis included percentages, mean values, standard deviations, and other parameters. The inferential analysis included Chi-square, independent *t*-test, one-way ANOVA, and post-hoc analysis. For testing the statistical hypothesis, results were considered statistically significant with a *p*-value lower than 0.05.

## 3. Results

### 3.1. Demographic Characteristics

A total of 144 participants joined this study, including 57 in the S group, 39 in the R group, and 48 in the C group. Their mean age was 27.82 years (SD = 7.79) in the S group, 24.74 years (SD = 7.09) in the R group, and 24.17 years (SD = 4.44) in the C group. Their mean education years were 14.04 years (SD = 2.48), 14.21 years (SD = 2.04), and 16.10 years (SD = 1.70), respectively. More than half of the participants were female, which was 29 (50.9%) in the S group, 24 (61.5%) in the R group, and 31 (64.6%) in the C group. The majority of the participants were single in terms of marital status (130, 90.3%). A total of 80 (55.6%) of them reported as non-religious. In the S group, 57 (100%) had previous mental health service-seeking experiences; in the R group, 20 (51.3%); and in the C group, 4 (8.3%). Lastly, the participants who reported family history of mental illness numbered 23 (40.4%), 15 (38.5%), and 7 (14.6%), respectively.

### 3.2. The Differences in Physiological Index, Lifestyles, and Physical Activity

Regarding the physical and physiological indices, the R group appeared to have higher levels in glucose (AC), triglyceride, and HDL than the average values; as well as lower scores in BMI, systolic BP, diastolic BP, and waist/hip ratio, albeit without statistical significance (Table 1).

The S group and the R group scored significantly lower than the C group in almost all lifestyle dimensions. However, the C group had better nutrition than the R group. In addition, the three groups were similar in health responsibility. As for physical activity, the R group had a higher amount of moderate aerobic exercise than the S group.

### 3.3. The Differences in Mental Risk among the Three Groups

The S group and the R group scored significantly higher for almost all variables of anxiety, schizotypal personality, putative pre-psychotic states, and prodromal symptoms than the C group. The exceptions were interpersonal deficits (where the R group scored higher than the S and C groups) and positive symptoms, negative symptoms, and SOPS total, in which the S group scored higher than the R and C groups (Table 2).

### 3.4. The Gender Differences in R Group

As for physical and physiological index, the female participants were significantly higher in HDL (*t* = 4.31, *p* = 0.00) and significantly lower in BMI (*t* = −3.86, *p* = 0.00), systolic BP (*t* = −7.84, *p* = 0.00), diastolic BP (*t* = −4.39, *p* = 0.00), waist/hip ratio (*t* = −6.49, *p* = 0.00), and triglyceride (*t* = −2.25, *p* = 0.03) than the male participants. They also had significantly more individuals who completed 150 min of moderate aerobic exercise per week compared to the male participants (Table 3).

In mental risk, the female participants in the R group scored significantly higher in self-depreciating descriptions (*t* = 2.35, *p* = 0.02) and positive symptoms (*t* = 2.35, *p* = 0.02) and significantly lower in nutrition (*t* = −2.42, *p* = 0.02) compared to the male participants (Table 4).

## 4. Discussion

This study compared the differences in mental risks, lifestyles, exercise level, physical measurements, and metabolic index among healthy individuals, patients with schizophrenia, and individuals with UHR in Taiwan. The results show that the individuals with UHR and the patients with schizophrenia scored very similarly in many dimensions. This is expected because UHR is nearly considered to be the prodromal stage of psychosis [9,20,31]. However, some critical differences were also found.

One of the differences was that the individuals with UHR had more severe interpersonal deficits than the patients with schizophrenia and the general public. Poorer quality of interpersonal relationships [32,33] and lower level of social functioning [34] for individuals with UHR were already reported by previous researchers. The interpersonal difficulties experienced by individuals with UHR may stem directly from their fear of their own mental symptoms and the associated social stigma [32]. This makes them conceal themselves, which impacts their interpersonal relationships and social functioning.

Although individuals with UHR continue to be in the same environment as before, their psychological wellbeing starts to be disturbed by current mental symptoms [9,10,20]. Their functional performance can be more impaired in the absence of proper medical care [4]. Many of them are no longer able to live up to the expected standards previously given by peers, family members, or themselves and thus experience higher levels of psychosocial stress and significant deficit in protective factors [35], which may also be a reason that causes their interpersonal relationships and social functioning to deteriorate [32].

It caught our attention that interpersonal deficits in the individuals with UHR were even worse than those in the patients with schizophrenia in this study. The patients with schizophrenia in this study had already received medical care and had been on antipsychotic medications for a few years. Their working performance expectations from the public might be lower due to the neurocognitive deficit caused by the illness [36]. Enhancing self-awareness and goal-setting capabilities may significantly improve the interpersonal relationships and social functioning of individuals, potentially surpassing those in the UHR group. Future interventions should therefore concentrate on developing the self-awareness, social skills, and communication abilities of individuals with UHR.

Nutrition was the only lifestyle factor that the individuals with UHR scored significantly lower on than the general public, and the patients with schizophrenia had no differences to the other two groups. The results of this study show that 66.7% to 94.9% of the individuals with UHR care less about preservative and additive intake, fiber intake, six categories of food intake, and three meals per day. Previous research has also demonstrated that the presence of mental illness is often associated with poorer dietary patterns, characterized by higher intakes of refined carbohydrates and total fats and lower intakes of omega-3 and omega-6 fatty acids, fiber, fruits, vegetables, and essential minerals and vitamins [37]. Unhealthy dietary intake has been observed in the early stages of psychosis [38]. Researchers have noted that individuals in the clinical high-risk phase for psychosis frequently exhibit dietary patterns characterized by low fiber intake and high saturated fat consumption [15,16]. Such dietary habits may contribute to poorer overall health outcomes in the future. Therefore, future research and novel interventions should focus on modifying the dietary constituents or patterns of individuals with UHR.

Many studies have already shown that individuals with psychotic symptoms have a higher tendency to be physically inactive [9,14,15,16]. The results of this study show that individuals with UHR had higher exercise levels than the other two groups. The results of this study contrast with findings from other studies [15,16], which reported a low proportion of individuals in the clinical high-risk phase for psychosis meeting the recommended guidelines for physical activity. This divergence might be attributed to the heightened levels of anxiety observed in individuals with ultra-high risk (UHR) in this study. Regular exercise is recognized as an effective stress management strategy [5,39] and was utilized by many UHR individuals as a coping mechanism to reduce anxiety. Therefore, continuing the strategy of exercise for stress management is highly recommended for future interventions.

As for gender differences, male individuals with ultra-high risk (UHR) exhibited higher levels of negative symptoms and substance abuse, while females demonstrated greater general psychopathology and were more frequently diagnosed with comorbid affective and anxiety disorders [40]. This study further found that the female individuals with UHR required additional attention to self-esteem, positive symptoms, and nutrition intake compared to the males. On the other hand, men had lower exercise levels than women. These results suggest that gender-specific interventions should target these areas to be beneficial in improving outcomes [41].

This study also had few limitations. First of all, like all cross-sectional research, this study is limited to one time of measurement. Furthermore, the cross-sectional design of the study precludes the establishment of causality between risk factors and mental health outcomes. To ascertain the temporal sequence of events and establish a clearer causal relationship, longitudinal studies are necessary.

The changes over a period of time were uncertain. Moreover, some patients with schizophrenia might have still been taking antipsychotics during the assessment and thus scored better results. Lastly, although some physical and blood serum assessments were included, this study was mainly composed of subjective self-reported questionnaires. Future research can include more precise and non-invasive objective physiological measures, such as EEG or MRI.

Since individuals with UHR and schizophrenia patients were likely referred by psychiatrists, and the healthy controls were recruited from the general public, there exists a potential for selection bias. This bias could significantly influence the observed differences, as these groups might inherently differ in aspects not considered by the study, such as access to healthcare and motivation for participation. Moreover, the findings from the specific demographic of 13–45 years old may not be generalizable to other age groups or populations with different cultural backgrounds or healthcare systems.

## 5. Conclusions

This cross-sectional study was designed to examine the differences in both physical and psychological health among healthy individuals, patients with schizophrenia, and individuals with UHR. Our findings demonstrate that the individuals with UHR were similar in many dimensions to the patients with schizophrenia. The exceptions were lower positive symptoms, lower negative symptoms, lower prodromal symptoms, higher interpersonal deficits, lower nutrition intake, and higher level of exercise than the patients with schizophrenia. Future research and clinical practice should consider these findings to precisely target the specific needs of individuals at UHR. We highly recommend adopting intervention strategies that include enhancing self-awareness, social skills, and communication abilities; modifying dietary constituents and patterns; and utilizing exercise as a stress management technique.

## Figures and Tables

**Table 1 behavsci-14-00395-t001:** The differences in physical index, lifestyles, and exercise among the three groups.

Variable	Total (*N* = 144)		S Group (*n* = 57)		R Group (*n* = 39)		C Group (*n* = 48)		*F*	*η* ^2^	Post Hoc Analysis
	Mean	SD	Mean	SD	Mean	SD	Mean	SD			
Physical Assessments											
BMI	23.71	4.84	23.69	4.95	23.26	5.11	24.09	4.55	0.31	0.004	
Systolic BP	114.71	16.24	116.56	17.20	111.46	15.98	115.15	15.19	1.17	0.016	
Diastolic BP	75.97	11.27	76.91	13.31	74.79	9.67	75.81	9.86	0.41	0.006	
Waist/Hip Ratio	0.84	0.07	0.85	0.08	0.83	0.07	0.82	0.06	2.12	0.029	
Serum Assessments											
Glucose (AC) (mg/dL)	86.38	23.83	84.73	29.90	91.92	11.30	83.90	23.02	1.43	0.020	
Triglyceride (mg/dL)	102.33	79.66	112.42	88.63	104.81	95.70	88.85	49.15	1.15	0.016	
HDL (mg/dL)	55.09	16.49	51.89	14.72	55.68	15.23	58.30	18.87	1.99	0.028	
Health Promotion Lifestyles											
Self-actualization	9.28	3.56	8.51	3.53	7.46	2.73	11.69	2.89	22.66 **	0.243	C > S, R
Health Responsibility	6.88	2.27	6.54	2.23	6.62	2.55	7.48	1.98	2.62	0.036	
Exercise	7.47	2.52	6.93	2.13	6.92	2.61	8.54	2.58	7.12 **	0.092	C > S, R
Nutrition	9.17	2.62	9.21	2.50	7.97	2.51	10.08	2.52	7.63 **	0.098	C > R
Interpersonal Support	10.17	3.07	9.04	2.59	8.79	2.75	12.63	2.28	33.83 **	0.324	C > S, R
Stress Management	9.26	2.56	8.60	2.07	8.00	2.33	11.08	2.29	25.14 **	0.263	C > S, R
HPLP-S Total	52.22	12.58	48.82	11.15	45.77	10.50	61.50	10.38	27.89 **	0.284	C > S, R
Physical Activities											
Moderate Aerobic Exercise	71.81	141.69	34.21	69.46	110.66	179.40	84.90	161.78	3.82 *	0.051	R > S

* *p* < 0.05; ** *p* < 0.01; BMI, body mass index; BP, blood pressure; AC, before meals; HDL, high-density lipoprotein; S, patients with schizophrenia; R, individuals with UHR; C, healthy control individuals. *η*^2^, eta squared.

**Table 2 behavsci-14-00395-t002:** The differences in anxiety and mental risks among the three groups.

Variables	Total (*N* = 144)		S Group (*n* = 57)		R Group (*n* = 39)		C Group (*n* = 48)		*F*	*η* ^2^	Post Hoc Analysis
	Mean	SD	Mean	SD	Mean	SD	Mean	SD			
Anxiety											
State Anxiety	45.16	13.54	50.32	12.62	52.21	9.96	33.31	8.42	44.82 **	0.389	S, R > C
Trait Anxiety	54.26	12.70	58.74	9.85	62.15	9.53	42.54	9.23	55.62 **	0.441	S, R > C
Schizotypal Personality											
Cognitive–Perceptual Deficits	3.83	2.24	4.44	2.03	5.03	1.88	2.13	1.72	30.25 **	0.300	S, R > C
Interpersonal Deficits	4.33	2.61	5.16	1.92	6.15	2.37	1.85	1.41	63.68 **	0.475	R > S > C
Disorganization	2.34	1.98	3.11	1.72	3.49	1.70	0.50	0.92	55.31 **	0.440	S, R > C
CSPQ-B Total	10.49	6.07	12.70	4.40	14.67	5.14	4.48	3.23	73.64 **	0.511	S, R > C
Putative Pre-psychotic States											
Interpersonal Difficulty/Social Anxiety Symptoms	2.75	1.75	3.37	1.63	3.72	1.30	1.23	1.08	44.61 **	0.388	S, R > C
Self-depreciating Descriptions	1.60	1.23	2.02	1.09	2.23	0.96	0.60	0.94	35.92 **	0.338	S, R > C
Negative Symptoms	1.35	1.18	1.96	0.93	1.85	1.06	0.21	0.54	62.97 **	0.472	S, R > C
Subthreshold Psychotic-like Experiences	1.93	1.50	2.49	1.32	2.85	1.06	0.52	0.85	58.72 **	0.454	S, R > C
BQSPS Total	7.63	4.88	9.84	3.82	10.64	3.41	2.56	2.48	85.03 **	0.547	S, R > C
Prodromal Symptoms											
Positive Symptoms	6.26	6.39	11.16	6.11	7.03	3.62	0.02	0.14	88.56 **	0.557	S > R > C
Negative Symptoms	8.67	8.32	14.60	7.27	10.87	5.71	0.13	0.39	95.44 **	0.575	S > R > C
Disorganization Symptoms	3.13	3.68	5.29	4.17	3.82	2.56	0.13	0.39	41.19 **	0.369	S, R > C
General Symptoms	5.60	4.57	8.36	3.50	8.08	2.82	0.46	0.90	133.27 **	0.654	S, R > C
SOPS Total	23.65	20.44	39.42	16.56	29.79	9.40	0.73	1.18	153.34 **	0.685	S > R > C

** *p* < 0.01; S, patients with schizophrenia; R, individuals with UHR; C, healthy control individuals; CSPQ-B, Chinese version of Schizotypal Personality Questionnaire-Brief; BQSPS, Brief Self-Report Questionnaire for Screening Putative Pre-Psychotic States; SOPS, Scale of Prodromal Symptoms. *η*^2^, eta squared.

**Table 3 behavsci-14-00395-t003:** The gender differences in physical index, lifestyles, and exercise among the prodromal group.

Variables	Female (*n* = 24)	Male (*n* = 15)		
	Mean (SD)	Mean (SD)	*t*	*p*
Physical Assessments				
BMI	21.27 (4.43)	26.45 (4.57)	−3.51	<0.01
Systolic BP	103.71 (10.44)	123.87 (15.70)	−4.40	<0.01
Diastolic BP	70.83 (8.91)	81.13 (7.33)	−3.75	<0.01
Waist/Hip Ratio	0.79 (0.06)	0.89 (0.06)	−4.86	<0.01
Serum Assessments				
Glucose (AC)	90.75 (11.94)	93.93 (10.20)	−0.83	0.41
Triglyceride	76.96 (57.48)	150.57 (127.05)	−2.04	0.06
HDL	62.44 (14.77)	44.08 (6.69)	4.38	<0.01
Health Promotion Lifestyles				
Self-actualization	6.83 (2.22)	8.47 (3.23)	−1.88	0.07
Health Responsibility	6.67 (2.73)	6.53 (2.33)	0.16	0.88
Exercise	6.92 (2.39)	6.93 (3.01)	−0.02	0.98
Nutrition	7.25 (2.15)	9.13 (2.67)	−2.42	0.02
Interpersonal Support	8.67 (2.68)	9.00 (2.95)	−0.36	0.72
Stress Management	7.63 (2.00)	8.60 (2.75)	−1.28	0.21
HPLP-S Total	43.96 (8.85)	48.67 (12.50)	−1.38	0.18
Physical Activities				
Moderate Aerobic Exercise	134.69 (176.91)	72.22 (182.67)	1.06	0.30

BMI, body mass index; BP, blood pressure; AC, before meals; HDL, high-density lipoprotein.

**Table 4 behavsci-14-00395-t004:** The gender differences in anxiety and mental risks among the prodromal group.

Variables	Female (*n* = 24)	Male (*n* = 15)		
	Mean (SD)	Mean (SD)	*t*	*p*
Anxiety				
State Anxiety	51.54 (9.40)	53.27 (11.05)	−0.52	0.61
Trait Anxiety	63.04 (9.64)	60.73 (9.49)	0.73	0.47
Schizotypal Personality				
Cognitive–Perceptual Deficits	4.96 (2.05)	5.13 (1.64)	−0.28	0.78
Interpersonal Deficits	6.54 (2.00)	5.53 (2.83)	1.31	0.20
Disorganization	3.38 (1.61)	3.67 (1.88)	−0.52	0.61
CSPQ-B Total	14.88 (4.90)	14.33 (5.67)	0.32	0.75
Putative Pre-psychotic States				
Interpersonal Difficulty/Social Anxiety Symptoms	3.79 (1.06)	3.60 (1.64)	0.40	0.69
Self-depreciating Descriptions	2.50 (0.83)	1.80 (1.01)	2.35	0.02
Negative Symptoms	1.88 (1.12)	1.80 (1.01)	0.21	0.83
Subthreshold Psychotic-like Experiences	2.96 (1.04)	2.67 (1.11)	0.83	0.41
BQSPS Total	11.13 (2.91)	9.87 (4.07)	1.04	0.31
Prodromal Symptoms				
Positive Symptoms	7.92 (3.96)	5.50 (2.38)	2.35	0.02
Negative Symptoms	10.29 (6.02)	11.86 (5.19)	−0.81	0.42
Disorganization Symptoms	4.08 (2.45)	3.36 (2.76)	0.84	0.41
General Symptoms	8.38 (3.15)	7.57 (2.17)	0.84	0.40
SOPS Total	30.67 (9.17)	28.29 (9.95)	0.75	0.46

CSPQ-B, Chinese version of Schizotypal Personality Questionnaire-Brief; BQSPS, Brief Self-Report Questionnaire for Screening Putative Pre-Psychotic States; SOPS, Scale of Prodromal Symptoms.

## Data Availability

These study data are deidentified participant data. The data that support the findings of this study are available beginning 12 months and ending 36 months following the article publication from the corresponding author, W-FM, upon reasonable request at lhdaisy@mail.cmu.edu.tw.

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
