# Peer review of "The Comparison of Lifestyles, Mental Risks, and Physical Indices among Individuals with Ultra-High Risk of Psychosis, Patients with Schizophrenia, and the General Public"

_behavsci, 2024, doi:10.3390/bs14050395_

Round 1

Reviewer 1 Report

Comments and Suggestions for Authors

Throughout the paper there is a mixture of APA citations (lines 65, 114, 123, 132, 140, 141, 148, 242, 267, and 277) and Vancouver citations. Two citation systems cannot be used in the same paper.

On the other hand, in line 242 the ampersand is used in the text when the citation system does not allow it (Byrne and Morrison...).

In the introduction some of the references are somewhat old and a more recent paper could be included.

Finally, the conclusions are too brief and not sufficiently explained.

Author Response

Response Letter (Manuscript ID: behavsci-2921450)

 To The Reviewer:

Thank you very much for taking time from your busy schedule to comment on our manuscript. Our responses to your suggestions and questions are summarized below, with reference to the appropriate pages in the text. Revisions in the text have also been highlighted in RED. We truly appreciate your thoughtful and constructive comments that help make this a better paper. We are very grateful to you for all the comments and insightful suggestions that help enhance both the quality and readability of our paper. Thank you.

Comments and Suggestions for Authors

  1. Throughout the paper there is a mixture of APA citations (lines 65, 114, 123, 132, 140, 141, 148, 242, 267, and 277) and Vancouver citations. Two citation systems cannot be used in the same paper. On the other hand, in line 242 the ampersand is used in the text when the citation system does not allow it (Byrne and Morrison...). In the introduction some of the references are somewhat old and a more recent paper could be included. Finally, the conclusions are too brief and not sufficiently explained.

Response:

Thank you for reviewer’s comment. We changed all the citations format to Vancouver citations and added new citations (references 15-16) to replace old one. Meanwhile, we added more detail in conclusion. (please see line 9-14).

Reviewer 2 Report

Comments and Suggestions for Authors

Dear Author

The reviewer has the following comments to make about the manuscript:

INTRODUCTION

1. Authors have discussed the benefits of early intervention for mental health problems. They are suggested to be more specific regarding the interventions used and the evidence supporting their effectiveness.  

2. Authors have provided previous studies on cognitive and social dimensions among individuals with UHR, patients with schizophrenia, and healthy controls; however, the reviewer feels that it does not provide a comprehensive review of the existing literature. Many recent literature needs inclusion, e.g.

Provenzani U, De Micheli A, Damiani S, Oliver D, Brondino N, Fusar-Poli P. Physical Health in Clinical High Risk for Psychosis Individuals: A Cross-Sectional Study. Brain Sci. 2023 Jan 12;13(1):128.  

De Micheli A, Provenzani U, Krakowski K, Oliver D, Damiani S, Brondino N, McGuire P, Fusar-Poli P. Physical Health and Transition to Psychosis in People at Clinical High Risk. Biomedicines. 2024; 12(3):523.  

3. The identification of a gap in the literature is clear; but the authors are also suggested to  to include the significance of addressing this gap.  

MATERIAL AND METHODS

1. Provide a rationale for why the given study design was chosen over others, how participants were assessed for eligibility, why these specific instruments were chosen or how they were validated for use in the study population.  

2. Elaborate on how potential confounding variables were controlled for or how missing data were handled.

 3. The author mentioned that results were considered statistically significant with a p-value lower than 0.5, likely a typographical error.

 4.   While psychiatrists referred individuals with UHR and patients with schizophrenia, healthy control individuals were recruited from the general public, which may have resulted in selection bias and affected the comparability of the groups. Also, how eligibility criteria of control were ascertained  ( Absence of schizophrenia or mood disorder )is unclear.

 RESULTS

1. Make sure the p-value is correctly mentioned below the tables.

2. Include information about the effect size in the table if possible.

Discussion

 1. Compare the study findings with existing literature on ultra-high risk (UHR) individuals and schizophrenia in more detail.  

2. Acknowledge potential confounding factors ( particularly related to design) that could influence the observed differences between groups.  

3. Explore more on the clinical implications of these findings(beyond  modifying diet or implementing gender-specific interventions) e.g. how these findings could inform specific clinical practice or specific intervention strategies  .

CONCLUSION

1. Clearly mention the implications of the findings for clinical practice or research. e.g. how these differences might impact diagnosis, treatment, or prognosis.

2. Provide any specific recommendations for future research or clinical practice based on the study findings. Authors have mentioned that future research should consider the differences observed between UHR individuals and patients with schizophrenia. They do not provide clear guidance on how these findings can inform targeted interventions or support services for UHR individuals.  

Author Response

Response Letter (Manuscript ID: behavsci-2921450)

 To The Reviewer:

Thank you very much for dedicating time from your busy schedule to review our manuscript and provide insightful feedback. Below, we summarize our responses to your comments and questions. All revisions have been highlighted in red in the text. We are immensely grateful for your constructive suggestions, which have significantly enhanced the quality and readability of our paper.

Comments and Suggestions for Authors

The reviewer has the following comments to make about the manuscript:

INTRODUCTION

  1. Authors have discussed the benefits of early intervention for mental health problems. They are suggested to be more specific regarding the interventions used and the evidence supporting their effectiveness.

Response:

Thank you for your suggestion. We have added specific benefits of the interventions used and included relevant references. Please refer to lines (57-60) and (77-79) in the revised manuscript.

  1. Authors have provided previous studies on cognitive and social dimensions among individuals with UHR, patients with schizophrenia, and healthy controls; however, the reviewer feels that it does not provide a comprehensive review of the existing literature. Many recent literature needs inclusion, e.g.

Provenzani U, De Micheli A, Damiani S, Oliver D, Brondino N, Fusar-Poli P. Physical Health in Clinical High Risk for Psychosis Individuals: A Cross-Sectional Study. Brain Sci. 2023 Jan 12;13(1):128.  

De Micheli A, Provenzani U, Krakowski K, Oliver D, Damiani S, Brondino N, McGuire P, Fusar-Poli P. Physical Health and Transition to Psychosis in People at Clinical High Risk. Biomedicines. 2024; 12(3):523.  

Response:

Thank you for highlighting this omission. We agree with your assessment and have now incorporated the following recent studies into our literature review.

  1. The identification of a gap in the literature is clear; but the authors are also suggested to include the significance of addressing this gap.

Response:

Thank you for pointing out this critical aspect. We have clarified the significance of addressing the gap, emphasizing the importance of refining care plans for UHR individuals to better meet their specific needs. Please see lines (83-84).

MATERIAL AND METHODS

  1. Provide a rationale for why the given study design was chosen over others, how participants were assessed for eligibility, why these specific instruments were chosen or how they were validated for use in the study population.

Response:

Thank you for your inquiry. We chose a cross-sectional design due to its ability to efficiently compare multiple groups at a single point in time, which is crucial for observing potential differences or similarities without long-term follow-up. This design is also quicker and more cost-effective compared to longitudinal studies. Eligibility for participants was meticulously assessed through structured processes, including referrals from psychiatrists and screenings in the community. Detailed criteria were used to ensure a consistent and reliable classification of participants. Please see lines (300-316) for a discussion of the study design limitations.

In this study, participants' eligibility assessment was conducted through a structured process to ensure that individuals in the ultra-high risk group (UHR group), patients with schizophrenia (S group), and healthy controls (C group) met specific criteria. Participants in R & S groups were referred by psychiatrists from the psychiatric clinic. Healthy control individuals were recruited from the general public within the same community, possibly through advertisements, community outreach, or general health screenings. Meanwhile, individuals had to meet specific criteria of Prodromal Syndrome, identified through clinical assessments or structured interviews like the Structured Interview for Prodromal Syndromes (SIPS). Ensuring no history of comorbid substance use disorders and no current diagnosis of schizophrenia, mood disorders, or other psychotic disorders as per DSM-5 criteria. Besides ensuring general health, this group was also matched by age and gender with participants in the other groups.

  1. Elaborate on how potential confounding variables were controlled for or how missing data were handled.

Response:

We addressed potential confounding factors by matching or stratifying participants based on age and gender. Missing data were handled by replacing missing values with the mean, median, or mode of available data, and multiple confirmations were made during recruitment to prevent the occurrence of missing data.

  1. The author mentioned that results were considered statistically significant with a p-value lower than 0.5, likely a typographical error.

Response:

Thank you for catching that error. We have corrected the typographical mistake regarding the p-value, changing it from 0.5 to 0.05.

  1. While psychiatrists referred individuals with UHR and patients with schizophrenia, healthy control individuals were recruited from the general public, which may have resulted in selection bias and affected the comparability of the groups. Also, how eligibility criteria of control were ascertained (Absence of schizophrenia or mood disorder) is unclear.

Response:

Thank you for pointing this out, the recruitment was based on the Criteria of Prodromal Syndrome and DSM-5.

  1. The method of recruitment for the control group from the general public as opposed to a clinical setting can introduce selection bias. Healthy controls might differ systematically from the clinical groups in ways other than their mental health status, such as socioeconomic factors, general health behaviors, or access to healthcare. We agree that this bias is a common challenge in psychiatric research, and addressing it explicitly involves acknowledging the limitations of the recruitment methods and considering their impact on the study’s findings. So in this study, the healthy-control subjects recruited by matching age and gender with subjects in the previous two groups. The exclusion criteria were: DSM-5 criteria for schizophrenia, mood disorders, or other psychotic disorders. We have included the inclusion and exclusion criteria (please see line 109-111).
  2. This study was conducting structured clinical interviews by a psychiatrist (also as Co-PI) to ensuring that all participants are assessed in a consistent manner to ensure that control group participants do not have schizophrenia or mood disorder.

RESULTS

  1. Make sure the p-value is correctly mentioned below the tables.

Response:

Thank you for bringing up this issue. We had changed it accordingly.

  1. Include information about the effect size in the table if possible. 

Response:

We appreciate this suggestion and have added the effect sizes (η²) in Tables 1 and 2 to better illustrate the differences among the groups.

Discussion

  1. Compare the study findings with existing literature on ultra-high risk (UHR) individuals and schizophrenia in more detail.

Response:

Thank you for bringing our attention to this issue. We have expanded our discussion to more thoroughly compare our findings with the existing literature on UHR individuals and schizophrenia. Please see lines (245-247, 261-265, 270-279, and 283-290).

  1. Acknowledge potential confounding factors (particularly related to design) that could influence the observed differences between groups.  

Response:

Since individuals with ultra-high risk and schizophrenia patients were likely referred by psychiatrists and the healthy controls recruited from the general public, there is a potential selection bias. This bias could influence the observed differences as these groups might inherently differ in ways not accounted for by the study (e.g., access to healthcare, motivation for participation).in addition, the cross-sectional nature of the study does not allow for establishing causality between the risk factors and mental health outcomes. Longitudinal studies would be required to confirm the temporal sequence of events and establish a clearer causal relationship. The findings from a specific demographic (13-45 years old) may not be generalizable to other age groups or populations with different cultural backgrounds or health care systems. We have discussed the potential selection bias arising from the different recruitment methods and the cross-sectional nature of the study, which limits causality inference. These limitations and their implications for the study's findings are now thoroughly explored in the limitations section (lines 300-316).

  1. Explore more on the clinical implications of these findings (beyond modifying diet or implementing gender-specific interventions) e.g. how these findings could inform specific clinical practice or specific intervention strategies.

Response:

Thank you for pointing this out, in this study we found three critical differences among the three groups: interpersonal relationships, nutrition intake, and physical activity. Our intervention strategies for interpersonal relationships are improving self-awareness, social skills, and communication skills; for nutrition intake are modifying diet constituents or dietary patterns; and for physical activity are using as stress management technique. We had added more details in discussion (please see line 293-295).

 CONCLUSION

  1. Clearly mention the implications of the findings for clinical practice or research. e.g. how these differences might impact diagnosis, treatment, or prognosis.

Response:

Thank you for pointing this out, we had included the implications of the findings in more detail. Findings might indicate that existing treatments do not adequately address certain symptoms or are less effective for particular groups. This can prompt the development of new therapeutic approaches or the adaptation of existing ones to better meet the needs of these populations (please see line (245-247, 261-265, 270-279, and 283-290).

  1. Provide any specific recommendations for future research or clinical practice based on the study findings. Authors have mentioned that future research should consider the differences observed between UHR individuals and patients with schizophrenia. They do not provide clear guidance on how these findings can inform targeted interventions or support services for UHR individuals.

Response:

Thank you for bringing this up. We had provided more specific recommendations for future research and clinical practice. Based on the study's findings, developing specialized support services that focus on interpersonal deficits, nutritional counseling, and encouraging physical activity can be beneficial. These services should be accessible and specifically designed to meet the unique needs of UHR individuals. Given the cross-sectional nature of the current study, longitudinal research is recommended to track the progression of symptoms and lifestyle changes over time in individuals atUHR of psychosis compared to schizophrenia patients and healthy controls. This can help in understanding the temporal relationship between early symptoms and the development of full-blown psychosis. Future research should consider designing and implementing intervention trials that are specifically tailored to the needs identified in UHR individuals, such as interpersonal skills training, nutritional guidance, and physical activity programs. The effectiveness of these interventions in preventing psychosis and improving overall well-being should be assessed (please see line 324-328).

Round 2

Reviewer 2 Report

Comments and Suggestions for Authors

Thanks for addressing the comment. Hope to cary forward futher.